# Comparative Analysis of Chloroplast Genome in *Saccharum* spp. and Related Members of ‘*Saccharum* Complex’

**DOI:** 10.3390/ijms23147661

**Published:** 2022-07-11

**Authors:** Sicheng Li, Weixing Duan, Jihan Zhao, Yanfen Jing, Mengfan Feng, Bowen Kuang, Ni Wei, Baoshan Chen, Xiping Yang

**Affiliations:** 1Guangxi Key Laboratory of Sugarcane Biology, Guangxi University, Nanning 530004, China; lamina0130@126.com (S.L.); fjnldxzxs@163.com (J.Z.); mengfanfeng@st.gxu.edu.cn (M.F.); kuangbowen97@163.com (B.K.); 1917301038@st.gxu.edu.cn (N.W.); 2State Key Laboratory of Conservation and Utilization of Subtropical Agro-Bioresources, Guangxi University, Nanning 530004, China; 3Sugarcane Research Institute, Guangxi Academy of Agricultural Sciences, Nanning 530007, China; duanweixing84@126.com; 4Ruili Breeding Station of Sugarcane Research Institute, Yunnan Academy of Agricultural Sciences, Ruili 678600, China; rljyf@126.com

**Keywords:** sugarcane, phylogenomics, InDel marker, dCAPS marker

## Abstract

High ploids of the sugarcane nuclear genome limit its genomic studies, whereas its chloroplast genome is small and conserved, which is suitable for phylogenetic studies and molecular marker development. Here, we applied whole genome sequencing technology to sequence and assemble chloroplast genomes of eight species of the ‘*Saccharum* Complex’, and elucidated their sequence variations. In total, 19 accessions were sequenced, and 23 chloroplast genomes were assembled, including 6 species of *Saccharum* (among them, *S. robustum*, *S. sinense*, and *S. barberi* firstly reported in this study) and 2 sugarcane relative species, *Tripidium arundinaceum* and *Narenga porphyrocoma*. The plastid phylogenetic signal demonstrated that *S. officinarum* and *S. robustum* shared a common ancestor, and that the cytoplasmic origins of *S. sinense* and *S. barberi* were much more ancient than the *S. offcinarum*/*S. robustum* linage. Overall, 14 markers were developed, including 9 InDel markers for distinguishing *Saccharum* from its relative species, 4 dCAPS markers for distinguishing *S. officinarum* from *S. robustum,* and 1 dCAPS marker for distinguishing *S. sinense* and *S. barberi* from other species. The results obtained from our studies will contribute to the understanding of the classification and plastome evolution of Saccharinae, and the molecular markers developed have demonstrated their highly discriminatory power in *Saccharum* and relative species.

## 1. Introduction

Sugarcane (*Saccharum* spp.), belonging to Poaceae, Panicoideae, Andropogoneae, and Saccharinae, is widely cultivated in tropical and subtropical regions in more than 90 countries (http://www.fao.org, accessed on 1 March 2022). The *Saccharum* genus consists of two wild species, *S. spontaneum* and *S. robustum*, and two ancient cultivars, *S. barberi* for cultivars from India and *S. sinense* from China, along with “noble” sugarcane *S. officinarum*, which has been used for sugar production for hundreds of years. Modern sugarcane interspecific hybridization work was undertaken in Java during the 1920s [1]. Since then, several founder sugarcane cultivars/hybrids have been frequently used as parental material for breeding, resulting in a narrow genetic base for modern sugarcane cultivars, which limits the continued improvement of commercial traits of varieties.

The ‘*Saccharum* complex’ has some genera closely related to sugarcane that are divergent in a large number of biological and agronomic traits but capable of interbreeding with sugarcane. Among them, *Tripidium arundinaceum* (2n = 60) [2] and *Narenga porphyrocoma* (2n = 30) [3] have excellent field characteristics, such as a high biomass, barren and drought tolerance, early maturity, strong stems, strong tillering ability, and mosaic disease resistance [4]. In particular, considerable efforts had been made in introgressing robust traits from *Tripidium* species into sugarcane. Meanwhile, *N. porphyrocoma* has also been used for sugarcane breeding through distant hybridization [3]. The F_1_ and BC_1_ hybrid of sugarcane and *N. porphyrocoma* showed resistance to smut [5]. Since the *Saccharum* complex/Saccharinae comprises gene resources that are helpful for sugarcane variety improvement, it is necessary to obtain an accurate phylogenetic relationship between *Saccharum* and its purported relatives, particularly for those species from outside of the core *Saccharum*.

Chloroplasts (cp) are inherited maternally in most plants, and each of these organelles contains a quadripartite circular molecule of double-stranded DNA that comprises two inverted repeats (IRs), and single-copy regions, the large and small single-copy regions (LSC and SSC). Like most angiosperms, the chloroplasts of sugarcane are inherited maternally [6]. As a result of their relatively small size, simple structure, and conserved gene content, cpDNA sequences have been widely used for phylogenetic studies, and complete cp genome sequences could provide valuable datasets for resolving taxonomical complexes and/or DNA barcoding (e.g., *matK* and *rbcL*) [7]. In addition, the cp genomes have proven to be more informative than cp DNA fragments in revealing the phylogeny of land plants [8,9]. With the advent of next-generation sequencing technologies, the complete chloroplast genome sequences obtained by high throughput techniques could significantly increase the phylogenetic resolution and provide new insight into the taxonomic relationships of the S*accharum* complex. Nevertheless, based on the results obtained recently, distinguishing closely related species using universal or specific DNA barcodes may be extremely difficult [10]. A comparative analysis of the complete sequences of chloroplast genomes was successfully used in the *Aldama* [11] and t*Pinus mugo* complex [12], and their candidate barcode regions were potentially useful for genetic identification. *Saccharum* species have been restricted to being investigated in nuclear genomics because of their high ploids and large genome size, whereas their chloroplast genomes are much smaller and relatively simple. Therefore, a comparison of the complete cp genome sequences could reveal novel genome features such as single-nucleotide polymorphisms (SNPs), insertions/deletions (InDels), and microsatellites. Moreover, SNPs-based dCAPS markers have been utilized for genetic analysis and selective breeding in allopolyploid species such as oats [13], wheat [14], and *Brassica napus* [15].

In *Saccharum*, the complete cp genomes of *Saccharum* hybrid NCo310 [16], SP80-3280 [17], and RB867515 [18]; *S. spontaneum* SES205A, SES234B [19], and Yunnan 83-184 [20]; *S. officinarum* Badila and IJ76-514 were obtained in previous studies. More recently, the complete cp genomes of *T. arundinaceum* [21,22] and *N. porphyrocoma* [23] have been sequenced and assembled. The analysis of chloroplasts in *Saccharum* and related species focused on three aspects: (1) the structural features of the cp genome were compared with *Saccharum* species and other agronomically and industrially important monocots, such as *Zea mays* and *Sorghum bicolor* [16]; (2) the molecular classification of all *Saccharum* species and their nearest evolutionary neighbors was elucidated by plastid-based studies [23]; (3) the plastid view on sugarcane origins [20]. These results have indicated the potential of this approach to elucidate the phylogenetic relationships within the ‘*Saccharum* complex’.

However, as the major member of *Saccharum* species, the cp genomes of *S. robustum*, *S. barberi*, and *S. sinense* have not been sequenced and assembled yet, which is urgently required for a comprehensive understanding of the genus *Saccharum*. Meanwhile, the cp genomes of representative sugarcane germplasm were absent. Molecular sequence studies on the *Saccharum* genus to date have been small in scale, involving either limited marker studies or being based only on phylogenies derived from a few chloroplast regions or a few genomic regions. The situation is even more complex when attempting to distinguish between cultivars that might only be a few decades divergent from one another. In this study, we aimed to obtain the complete cp genome sequences of *Saccharum* and its relative genera to characterize those genome structures, gene content, and other characteristics. The whole-genome sequencing (WGS) paired-end reads were used to assemble the corresponding plastomes to (1) enrich, update, and improve the database genomics of Saccharinae, (2) improve the phylogenetic resolution of the defined species complexes, and (3) develop efficient markers for germplasm evaluation and species discrimination in *Saccharum* and *Saccharum*-relative species.

## 2. Results

### 2.1. Plastid Genome Structure and Organization

We sequenced and de novo assembled 19 cp genomes of Saccharinae members. Combined with 4 genomes that were assembled using raw reads from NCBI SRA database, a total of 23 chloroplast genomes (Appendix A) was used for plastid genome comparison, which included one species from *Tripidium*, one from *Narenga*, and twenty-one species from *Saccharum* (four *S. spontaneum* species, four *S. robustum* species, two *S. sinense* species, three *S. barberi* species, four *S. officinarum* species, and four *S.* hybrids). The complete chloroplast genome of *N. porphyrocoma* (141,231 bp) (Figure 1a) contained a large single-copy (LSC) region (83,107 bp), and a small single-copy (SSC) region (12,533 bp), which were separated by a pair of IRs (IRa and IRb; 22,794 bp each), 8 rRNA, and 38 tRNA, whereas that of *T. arundinaceum* (141,209 bp) (Figure 1b) consisted of an LSC (83,167 bp), an SSC (12,513 bp), and a pair of IRs (22,763 bp each), 8 rRNA and 38 tRNA.

In *Saccharum*, assembly sizes ranged from 141,161 bp (IND81-013, *S. spontaneum*) to 141,216 bp (SP79-9, *S. spontaneum*) (Figure 1c). All had a typical quadripartite structure, comprising an SSC (12,538–12,542 bp) and an LSC (82,036–83,093 bp) region separated by a pair of IRs (22,789–22,795 bp). The cp genome length varied from 141,161 to 141,216 bp, with a medium-size of 141,177 bp. In *Saccharum*, *S. spontaneum* ($\bar{x}$ = 141,183 ± 26 bp) had the most variable length, followed by *S. robustum* (141,171–141,187 bp, $\bar{x}$ = 141,179 ± 9 bp) and *S. officinarum* (141,182–141,183 bp, $\bar{x}$ = 141,183 ± 1 bp). Across the 21 chloroplast genome sequences in *Saccharum*, the cp genome length was most conserved in *S. sinense* (141,174 bp, $\bar{x}$ = 141,174 ± 0 bp) and *S. barberi* (141,174 bp, $\bar{x}$ = 141,174 ± 0 bp), as well as *S.* hybrid (141,182 bp, $\bar{x}$ = 141,182 ± 0 bp) (Table 1).

In terms of synteny and gene number, both were highly conserved in the assembled accessions in *Saccharum*, *Tripidium*, and *Narenga*. Gene annotation showed that there were 130 genes in all assembled cp genomes, including 83 protein-coding genes (74 single copy protein-coding genes, 7 repeated/IR genes, and 2 copies of *ndhB*, *rps12*, *rps19*, *rpl2*, *rpl23*, *rps7*, and *rps15*), 38 tRNA genes (9 repeated/IR genes and 2 copies of *trnA-UGC*, *trnH-GUG*, *trnI-CAU*, *trnV-GAC*, *trnI-GAU*, *trnA-UGC*, *trnR-ACG*, *trnN-GUU*, and *trnL-CAA*), and 8 rRNA genes (4 repeated/IR genes and 2 copies of *rrn23*, *rrn4.5*, *rrn5*, and *rrn16*) (Appendix A). Among the annotated genes, 13 genes had introns in all species, of which, there were two introns for *ycf3* and one intron for the other 12 genes (*atpF*, *ndhA*, *ndhB*, *rps16*, *rpl16*, *rpl2*, t*rnK-UUU*, *trnS-CGA*, *trnL-UAA*, *trnV-UAC*, *trnT-CGU*, and *trnA-UGC*). Interestingly, the gene *rps12* had two introns but with a trans-splicing, with one of its exons located in the LSC region, and the other duplicated by the IRs. The members of the *Saccharum* complex lost *accD* (encoded subunits of Acetyl-CoA carboxylase) and *ycf1*, which were also absent in the cp genomes of other Panicoideae grasses [16].

### 2.2. Phylogenetic Analysis of Saccharinae Species

Maximum likelihood (ML) phylogenetic analysis was performed based on 74 protein-coding genes (PCGs), which were highly conserved single copy protein-coding genes shared among 36 species in Saccharinae (23 cp genomes obtained in this study) (Appendix A), rooted using *Sorghum bicolor* as outgroup. The phylogenetic tree showed that all *Saccharum* species were clearly clustered into a lineage (Figure 2). *S. spontaneum* diverged from other *Saccharum* species firstly, whereas *S. sinense* and *S. barberi* were clustered into the same clades, being sisters to *S. robustum* and *S. officinarum**,* and they probably had different *S. officinarum* accessions as an ancestor in the hybridization, contributing to their differences in genome compositions. The phylogram showed that *S. officinarum* and sugarcane modern cultivar were closer to *S. robustum* than ancient hybrid species *S. sinense* and *S. barberi*.

### 2.3. Repeat Analysis and Genetic Diversity Assessment 

Chloroplast repeats with a high diversity in copy numbers are usually useful for markers development to investigate the population genetics and biogeography of allied taxa. In the MISA analyses, a total of 36 microsatellites regions (>=10 bp) were identified in the six *Saccharum* plastid genomes: 34 in *T. arundinaceum* and 33 in *N. porphyrocoma*. The distribution, types, and numbers of repeats were highly similar among the plastomes. The most abundant SSRs were mononucleotide, and the number of that was 26 in *T*. *arundinaceum* and *S. spontaneum*, and 27 in other species. The number of dinucleotide SSRs was the same in all species. Interestingly, trinucleotide SSRs were found in the plastomes of *T. arundinaceum*, and hexanucleotide SSRs were found in the plastomes of *N. porphyrocoma*, but were not detected in the six *Saccharum* plastomes. Detailed information about SSR types, SSR sequences, regions, and start/end positions of the aforementioned SSRs in the cp genome is presented in Appendix A.

There were no large structural variations, except for SNPs, and InDels in *Saccharum* cp genomes (Appendix A). Among six *Saccharum* species, *S. spontaneum* had the largest number of SNPs (68) and InDels (43), whereas *S. officinarum* had the smallest number of SNPs (10) and InDels (5). *S. robustum* had 20 SNPs and12 InDels that were detected, and *S. sinense* and *S. barberi* had the same number of SNPs (22) and InDels (10). Most of the SNPs were found in intergenic sequences (52), and the maximum number of SNPs was found in *ndhF-rpl32*(8), followed by *ndhC-trnV-UAC* (7), *psbM-petN* (4), *trnG-GCC-trnM-CAU* (4), *rps16-trnQ-UUG* (3), and *petA-psbJ* (3). Seven SNPs were reported from introns of six genes, which was *rps16* (1), *trnG-UCC* (1), *ycf3* (1), *trnV-UAC* (1), *rps12* (2), and *rps16* (1). Fifteen SNPs were located in the coding sequence (CDS) of genes involved in a variety of biological processes, from transcription (*rpoC1* and *rpoC2*) to translation (*rps8* and *rps3*), the subunits of ATP synthase (*atpA*, *atpB*, and *atpF*), subunits of photosystem II (*psbE* and *ycf3*), c-type cytochrome synthesis (*ccsA*), and subunits of NADH-dehydrogenase (*ndhF* and *ndhD*). Mostf InDels were found in intergenic sequences, except for two InDels found in *ycf3* introns in *S. spontaneum*. Two long insertion fragments found in *S. spontaneum* were 27 bp and 50 bp, whereas one base insertion was found in other *Saccharum* species.

### 2.4. Comparison of Chloroplast Genome Sequences

The LSC/IRs and IRs/SSC borders and their adjacent genes were compared in the six *Saccharum* cp genomes and other Saccharine species (Figure 3). The *rps19* gene was located in the IRb region entirely, 35 bp away from the LSC/IRb border. The *ndhF* gene was situated at the junction of IRb/SSC, extending 29 bp into the IRb region. Overall, the structure and gene content of the six chloroplast genomes were consistent, and no significant expansion or contraction of IR regions were found in the *Saccharum* spp. and two related species.

To investigate the similarities and differences in the cp genome sequences between *Saccharum* and other species of Saccharinae, a global alignment program was used to align these sequences. The result was plotted using the mVISTA tool with *sorghum bicolor* as a reference. Those closely related species had little difference in cp genome size, ranging from 141,174 bp to 141,231 bp. As shown in Figure 4, the structures of these cp genomes were conserved. The global patterns of sequence similarities among these sequences were very high.

A comparative sequence analysis of the eight Saccharinae species using mVISTA revealed a high sequence identity, with identity ranging from 97.89% to 99.95%, suggesting that Saccharinae chloroplast genomes were extremely conserved. Overall, the tRNA region was the most conservative (identity = 99.95 ± 0.01%), and exon regions (identity = 99.80 ± 0.00%) were more conserved than the intron region (identity = 99.04 ± 0.14%) and intergenic regions (identity = 98.19 ± 0.19%). Most high polymorphic regions were located in the intergenic regions (such as *rps16-trnQ*, *trnG-trnM*, *psbM-petN*, *ndhC-trnV*, *petA-psbJ*, and *psbJ-rpl33*). These regions may be undergoing more rapid nucleotide substitution at the species level, indicating their potential applications for developing molecular markers for phylogenetic analyses and plant identification in Saccharinae species. A few divergent regions were also observed in the genes coding regions of *matK*, *psbB*, *psaB,* and *rpoA*, and introns of *atpF*, *rps16*, *rpl16*, and *ycf3*.

### 2.5. Development of Markers to Discriminate Saccharum and Sugarcane-Relative Species

The nucleotide diversity (Pi) of the chloroplast genomes in the six *Saccharum* species (including 21 accessions) was also calculated to assess the sequence divergence level of these species (Figure 5). The average Pi value was 0.000265 (range 0–0.00518) in the LSC region, and 0.000468 (range 0–0.00324) in the SSC region, whereas that of the IR region was 0. Most of the sequences with high Pi values were spacer regions between genes. Among these spacer regions, *rps16-trnQ*, *trnG-trnM*, *psbM-petN*, *petN-trnC*, *ndhC-trnV*, *petA-psbJ*, *psbJ-rpl33*, *ndhF-rpl32*, and *trnL-ccsA* were the nine regions with Pi values bigger than 0.01000. Only one coding region (*ycf3*) had high Pi values over 0.01000. These hotspot regions of divergence could be applied in developing DNA markers for distinguishing *Saccharum* species. We also calculated Pi for chloroplast genome sequences of each species to investigate the sequence variation at interspecies level. The highest Pi value was 0.00007 in *S. robustum*, followed by that of *S. spontaneum* with 0.00003, whereas the lowest Pi value was 0 in *S. officinarum*, *S. sinense*, *S. barberi*, and *S. hybrids*. The Pi values of the two wild species (*S. spontaneum* and *S. robustum*) within the genus *Saccharum* were higher than that of the domesticated species (*S. officinarum*, *S. sinense*, *S. barberi*, and *S. hybrids*), showing that the wild species had the higher sequence diversity.

Based on the alignment of complete cp genome sequences, the nine most variable loci based on the analysis of all twenty-one individuals were selected as DNA markers for development (Figure 6a). Markers were derived from InDels in the intergenic regions, including *rps16-trnQ*, *trnS-psbD*, *trnS-psbZ*, *trnR-rps14*, *ndhC-trnV*, *petL-petG*, *rpl33-rps18*, and *rps12* introns (Table 2). These nine markers were classified into three categories based on their differentiation ability. For the first category, SscpId4 in the *trnR-rps14* intergenic region was able to differentiate each of the three species, *T. arundinaceum*, *S. spontaneum*, and *S. officinarum*, which had a 188 bp, 172 bp, and 198 bp expected amplified fragment size, respectively. For the second category, the marker SscpId7 (15 bp InDel in the *rpl33-rps18* region) could differentiate *S. spontaneum* and *T. arundinaceum* from other *Saccharum* species. For the third category, all of the rest markers could differentiate *T. arundinaceum* from *Saccharum* species, which had a 29 bp, 31 bp, 34 bp, 21 bp, 46 bp, 18 bp, and 33 bp fragment length difference for SscpId1, SscpId2, SscpId3, SscpId5, SscpId6, SscpId8, and SscpId9, respectively. Regarding validation experiments using six samples, two accessions from each species showed the power of these markers to separate the target samples (Figure 6b). These markers were valuable tools for accurate germplasm evaluation in sugarcane research and breeding programs.

The species-specific SNPs between *S. robustum* and *S. officinarum* were used to develop dCAPS markers to generate the mismatches near SNPs for the recognition site of restriction endonuclease (Table 3). One SNP in *trnG-trnM* intergenic regions (“G” for *S. officinarum*, and “A” for *S. robustum*) was used to design two dCAPS markers Sscpd1 and Sscpd2 (Figure 7a), with the *TaqI* and *XbaI* digestion site in the amplicon fragment using *S. robustum* DNA as a template, respectively. Sscpd3 and Sscpd4 were designed for SNPs in the *petA-psbJ* (“G” for *S. officinarum*, and “A” for *S. robustum*) and *rbcL-psaI* (“C” for *S. officinarum*, and “G” for *S. robustum*) intergenic region, with the *TaqI* digestion site in the amplicon fragment using *S. officinarum* DNA as a template. The validation showed that the markers could separate *S. officinarum* (Badila) and *S. robustum* (NG57-208) clearly by the fragment length difference (Figure 7b), indicating that these dCAPS markers worked.

The species-specific SNPs between *S. sinense*, *S. barberi*, and other *Saccharum spp*. (including *S. hybrid*, *S. officinarum*, *S. robustum*, and *S. spontaneum*) were used to develop the dCAPS marker (Table 4). Sscpd5 designed for SNPs in the *rbcL-psaI* (“A” for *S. sinense* and *S. barberi*, and “G” for other *Saccharum* spp.) intergenic region, with the *TaqI* digestion of the amplicon producing a digested fragment, except in *S. sinense* and *S. barberi*, resulting in a 201 bp fragment size in *S. sinense* and *S. barberi*, and 216 bp in other *Saccharum* species, respectively. The validation results showed that these markers could clearly separate ancient cultivars *S. sinense* and *S. barberi* from modern sugarcane hybrids, *S. officinarum* and *S. robustum*, by the fragment length difference (Figure 8b), indicating that the dCAPS marker worked for the discrimination of *S. sinense/S. barberi* from other *Saccharum* species.

## 3. Discussion

In this study, we firstly sequenced and assembled cp genomes of *S. barberi*, *S. sinense*, and *S. robustum*. A total of 23 cp genomes of Saccharinae were presented, including 6 *Saccharum* species and 2 *Saccharum*-related species, *N. porphyrocoma* and *T. arundinaceum*, which enriched and improved the chloroplast database of Saccharinae. The chloroplast sequences of *Saccharum* species showed high variation in their intergenic regions. Based on sequence polymorphisms of chloroplast genomes assembled in this study, nine InDels markers and five dCAPS markers were successfully developed. Those markers not only enriched the marker for species discrimination in Saccharinae, but also had highly discriminatory power in *Saccharum* species and *Saccharum*-relative species.

### 3.1. Differences in Gene and Structure of Chloroplast Genomes among Saccharinae Species 

For the first time, the chloroplast genomes of the representative six species in *Saccharum* and two relative species *T. arundinaceum* and *N. porphyrocoma* have been described, which provides important insights into the characteristics of plastid genomes of the members of Saccharinea. Our results showed that the chloroplast genomes of *Saccharum* species were extremely similar, and the genome sizes were similar to those of most plant cp genomes reported previously [20,23,24,25,26], indicating their high level of conservation, especially in terms of chloroplast genome organization, gene content, and SSRs. The expansion and contraction of IR regions were the main cause of variation in cp genome sizes [27]. We compared the cp IR boundary regions of reported plastomes to those of closely related species and found slight differences between closely related species. The six chloroplast sequences of *Saccharum* species showed higher variation in non-coding regions than in their coding regions. Some of the mutation hotspot regions, including *rps16-trnQ*, *trnG-trnM*, *psbM-petN*, *petN-trnC*, *ndhC-trnV*, *petA-psbJ*, *psbJ-rpl33*, *ndhF-rpl32*, and *trnL-ccsA*, were detected from the cp genomes, which were potential regions for marker development for species discrimination.

### 3.2. Plastid View on the Evolutionary Relationships within Saccharinae

Within *Saccharum*, the lineage of *S. spontaneum* diverged from the other *Saccharum* species firstly, and *S. officinarum* diverged from a common ancestor with *S. robustum*. *S. sinense* and *S. barberi* were clustered as a monophyletic group with the *S. officinarum* and *S. robustum* linage, suggesting that their maternal origins might be the common ancestors. *S. sinense* and *S. barberi* were more distant to *S. robustum* than *S. officinarum* and modern sugarcane cultivar. As expected, high polymorphisms within species were observed in the *S. spontaneum* and *S. robustum* clade, possibly because they were wild species and had widely ecological distributions. Low polymorphisms were observed in branches *S. sinense* and *S. barberi*, indicating their limited maternal origins. 

The existing consensus within *Saccharum* was that *S. officinarum* originated from *S. robustum*, whereas *S. sinense* and *S. barberi* were believed to have arisen by the local/natural hybridization of *S. officinarum* with *S. spontaneum*. Recent studies [28] based on homologous haplotypes suggested that *S. officinarum* was presumed to be derived from *S. robustum* 0.64 Mya ago, and *S. officinarum* and *S. robustum* were derived from interspecific hybridization between two unknown ancestors [19]. The phylogram presented in the current study confirmed that *S. officinarum* shares a common ancestor with *S. robustum*, but the true natural evolutionary processes are still unclear. Our results demonstrated that *S. sinense* and *S. barberi* might be from a single-origin natural hybridization and might have arisen earlier than previously inferred (4000 years BP) [19]. However, more evidence is required to corroborate this conclusion. In addition, in the case of some closely related taxa, the use of cp genomes may fail to distinguish these taxa at the intrageneric level [11,12].

Within the ‘*Saccharum* complex’, *T. arundinaceum* and *N. porphyrocoma* were considered as excellent gene resources for sugarcane variety improvement. Many closely related species have been employed within sugarcane breeding programs, but the information on the phylogeny of the core ‘*Saccharum* complex’ is still limited, especially species such as *Erianthus rockii* Keng, a close relative that is used in the introgressing process with sugarcane. Previous studies had suggested that *Tripidium* and *Saccharum* were divergent eight million years ago, last sharing a common ancestor twelve million years ago [21]. However, the phylogram in this study showed that *T. arundinaceum* was sister to *E. aurea*, which was closely related to Miscanthus. The distant plastid phylogenetic relationship between sugarcane and *Tripidium* suggested that these taxa may be less cross-compatible with sugarcane than *Erianthus* and Miscanthus, the same as a previous study [25], and were therefore not ideal for crosses with sugarcane. Recently, a comparison of chloroplast genomes and ITS phylogenies revealed that *Erianthus* is monophyletic with Narenga and *Saccharum* [23], but the phylogram in this study revealed that *T. arundinaceum* (once named *Saccharum* arundinaceum [29] or *Erianthus arundinaceum* [22]) is clustered into the lineage of *Tripidium*. Therefore, these accessions should be classified as genus ‘*Tripidium*’ instead of its traditional genus ‘*Saccharum*’ or ‘Erianthus’. The phylogenomic analysis had a good boostraps branch that supported that *N. porphyrocoma* typically emerged as sister to *Saccharum*, demonstrating that Narenga was more closely related to *Saccharum* than *Tripidium*.

### 3.3. Development of Markers at the Inter- and Intra-Species Level for Saccharum

The cytoplasm plays a significant role in the inheritance and realization of agronomically important traits. To date, many examples of the influence of the cytoplasmic background on numerous plant traits, including those of great biological and economic importance, have been documented [30]. The identification of correct parental germplasm and verifying the cytoplasm origin of their progeny are crucial for sugarcane breeding. The matrilineal inheritance of chloroplasts makes it possible to use their SNPs for the identification of maternal parents [31].

Previously, researchers attempted to utilize cytogenetic tools such as GISH [32] or the detection of the progeny of hybrid populations from intra- or inter-species cross [33,34,35], and the identification of *Saccharum* germplasm materials. However, GISH is still very expensive, laborious, and time-consuming. Instead, species-specific molecular markers are favored to be utilized. At the beginning, instance ribosomal restriction fragment length polymorphism (RFLP) markers were used to distinguish *S. sinense* and *S. barberi* accessions from other *Saccharum* species, but their detectable variations were limited within species [36]. Then, PCR-based markers, such as the random amplification of polymorphic DNA (RAPD) markers and simple sequence repeats (SSR) or microsatellite markers, were applied. RAPD markers were used to identify “*Saccharum officinarum × Erianthus fulvus”* true F_1_ hybrids [37], but they proved to be less reproducibility across laboratories. The SSR marker is considered to be one of the most efficient markers, and has been utilized greatly to identify true hybrids of sugarcane [38,39,40,41,42,43]. Nevertheless, SSRs have some disadvantages over SNPs/InDels-based markers, such as a low mutation rate.

The large and polyploid genome of sugarcane is a challenge for markers development, but the cp genome provides a useful target for developing DNA markers to verify species. For the development of DNA markers using sequence variations between species with similar genetic backgrounds, it would be efficient to use SNPs and InDels. dCAPS markers generated from SNPs have been proven to have important applications in the analyses of genetic polymorphism and phylogeny in closely related species [31,44]. To date, no SNPs or InDels markers for species discrimination have been developed in the sugarcane cp genome. In this study, we successfully developed 14 valuable and unique polymorphic markers to distinguish *Saccharum* and related species. Most of these markers were derived from the intergenic regions of cp genomes and showed high interspecific polymorphisms. At the level of genus species differentiation, eight InDels-based markers could discriminate *T. arundinaceum* from *Saccharum* species. At the level of species differentiation within the genus *Saccharum*, two InDel markers could discriminate *S. officinarum* from *S. spontaneum*, four SNP-based dCAPS markers could discriminate *S. officinarum* from *S. robustum*, and one dCAPS marker could differentiate *S. barberi* and *S. sinense* from other *Saccharum* species. Those markers were fragment-length-difference-based (InDels) or SNP-transformed dCAPs, which are easy and cheap to run for sample discrimination. In addition, these markers have a good repeatability, and it is easy to transfer them between sugarcane populations or laboratories. It is possible to utilize these markers for germplasm evaluation and species discrimination in *Saccharum* and relative species.

## 4. Materials and Methods

### 4.1. DNA Extraction, Library Construction, and Sequencing

Fresh leaves of 19 samples were collected from Guangxi University (Nanning, Guangxi, China) (108°19′ E, 22°51′ N, 530,023) (Appendix A) and stored in liquid nitrogen. After the DNA extraction from fresh leaf tissues, its quantification was validated using Agarose gel electrophoresis and Nanodrop for construction of DNA library, and it was sequenced by Illumina NovaSeq 6000 sequencing platform (Illumina, San Diego, CA, USA). Approximately 10 Gb of 150 bp paired-end reads were generated for each accession.

### 4.2. Plastome Assemblies and Annotation

Fastp (v0.21.0) [45] was used for quality control, adapter trimming, quality filtering, and per-read quality pruning. The trimmed reads were subjected to GetOrganelle (v1.7.5) [46] to generate plastome assemblies as suggested for Embryophyta plant plastome sequences (-R 10; -F embplant_pt). The SPAdes [47] kmer settings were set to -k 21, 45, 65, 85, or 105. The contig coverage information and other graph characteristics were used by GetOrganelle to construct the final assembly graphs, which were plotted and visually assessed using Bandage (v0.8.1) [48]. The completed chloroplast assembly was re-aligned to *S. officinarum* cultivar badila (GenBank: MG685915.1) to confirm the quadripartite structures, including LSC, SSC, and two IRs.

Genome annotation and visualization were performed using web server CPGAVAS2 [49], CDS, and rRNA annotation based on BLAST search, while tRNAscan-SE v2.0.2 [50] was used to find and annotate tRNA genes. Then, the genomes were manually corrected using the Apollo [51]. Lastly, Chloroplot [52] was used to generate a corrected cp circular map.

### 4.3. Phylogenetic Analyses

For phylogenetics analysis for Saccharinae species, 14 published plastome sequences were downloaded. Sorghum bicolor (NC_008602.1) was used as outgroup. Fasta files with the nucleotide sequences for all 74 PCGs that they were shared from were extracted from the GenBank files using PhyloSuite (v1.2.2) [53]. All of the genes were aligned in batches with MAFFT (v7.313) [54] integrated into PhyloSuite using normal-alignment mode. The best partitioning scheme and evolutionary models for 74 pre-defined partitions were selected using PartitionFinder2 [55], with greedy algorithm and AICc criterion, which were divided into 27 subsets. PhyloSuite was then used to concatenate these alignments into a single alignment and generate phylip and nexus format files for the phylogenetic analysis. The selection of the best-fit partition strategy and models was carried out using PartitionFinder2. Maximum likelihood phylogenies were inferred using IQ-TREE [56] under Edge-unlinked partition model for 50,000 ultrafast [57] bootstraps, as well as the Shimodaira-Hasegawa-like approximate likelihood ratio test [58]. Images were drawn with iTOL [59] and Adobe Illustrator.

### 4.4. Plastome Polymorphism

To explore the sequence divergence and hotspot regions, DnaSP (v6.0) [60] was used for sliding window analysis for computing nucleotide diversity (pi) among the chloroplast genome sequences, with 500 bp windows size and 250 bp step size.

For variation analysis, clean reads were aligned to *S. officinarum* cultivar badila (GenBank: MG685915.1) cp genome v3.0 using BWA-mem [61] with default settings. For variant calling, we used the GATK HaplotypeCaller v.4.1.8.1 [62] software with the ‘-ploidy 1’ parameter to compare SNPs and small indels from BAM files. We filtered out low-quality variants (“QD < 2.0 || MQ < 40.0 || FS > 60.0 || SOR > 3.0 || MQRankSum < −12.5 || ReadPosRankSum < −8.0”) in GATK.

The microsatellites were determined by perl script MIcroSAtellite (MISA) [63] with search parameters: 1–10; 2–5; 3–5; 4–4; 5–3; 6–3, where 1, 2, 3, 4, 5, and 6 indicate the mono- di-, tri-, tetra-, penta-, and hexa-nucleotide repeats.

### 4.5. Comparative Genome Alignment

Aligned sequences and annotations for assembled chloroplast genomes were used to construct sequence conservation plots in the program mVISTA [64] and were compared using mVISTA software in Shuffle-LAGAN mode with the S. bicolor as a reference. IRscope [65] was used to detect IR border expansion and contraction.

### 4.6. Development and Validation of InDel and dCAPS Markers

To discriminate among the Sacchariane species, specific primers were designed using Primer 3 based on the mutational hotspot regions found in *Saccharum* and *Saccharum*-related chloroplast genomes. The specific primers for InDel regions and dCAPS primers for SNP sites were designed using the Primer 3 program (http://bioinfo.ut.ee/primer3-0.4.0/, accessed on 15 January 2022) and dCAPS Finder 2.0 (http://helix.wustl.edu/dcaps/dcaps.html, accessed on 15 January 2022), respectively.

Polymerase chain reaction (PCR) was carried out in a 20 μL reaction mixture containing 10 μL of 10× reaction buffer, 5 pmol of each primer, 1.25 units of Taq DNA polymerase, and 20 ng of DNA template. The PCR reaction was performed in thermocyclers using the following cycling parameters: 94 °C (5 min); 35 cycles of 94 °C (30 s), 56–62 °C (30 s); 72 °C (30 s), then 72 °C (7 min). The PCR products of each dCAPS marker were treated with a restriction enzyme at 37 °C (XbaI) or 65 °C (TaqI) for 1 h. PCR products were visualized on agarose gels (2.0–3.0%) containing safe gel stain.

## 5. Conclusions

In this study, we sequenced 19 accessions of Saccharinae species, and a total of 23 chloroplast genomes were assembled, including 6 species of *Saccharum* (among them, *S. robustum*, *S. sinense*, and *S. barberi* firstly reported in this study) and 2 sugarcane relative species, *Tripidium* arundinaceum and Narenga porphyrocoma. Our results enriched, updated, and improved the database genomics of Saccharinae, further confirming the phylogeny of Saccharinae based on available cp genomes. These results can be used to elucidate evolutionary history, such as the origin of Saccharinea, allowing for a deeper understanding of their parental lineages.

In addition, we developed 14 robust molecular markers, including 9 InDels markers for the discrimination of *S. spontaneum*, *S. officinarum*, *T. arundinaceum*, 5 dCAPS markers for *S. officinarum* and *S. robustum*, and 1 dCAPS marker for differentiated *S. sinense* and *S. barberi* from other *Saccharum* species. These markers were demonstrated to have a high discriminatory power in the *Saccharum* genus. Taken together, the results obtained from our study will contribute to the understanding of Saccharinae classification and plastome evolution at inter- and intra-species levels of *Saccharum*. Molecular markers developed in this study were inexpensive and efficient, and, thus, were possible to utilize for germplasm evaluation and species discrimination in Saccharum and relative species.

## Figures and Tables

**Figure 1 ijms-23-07661-f001:**
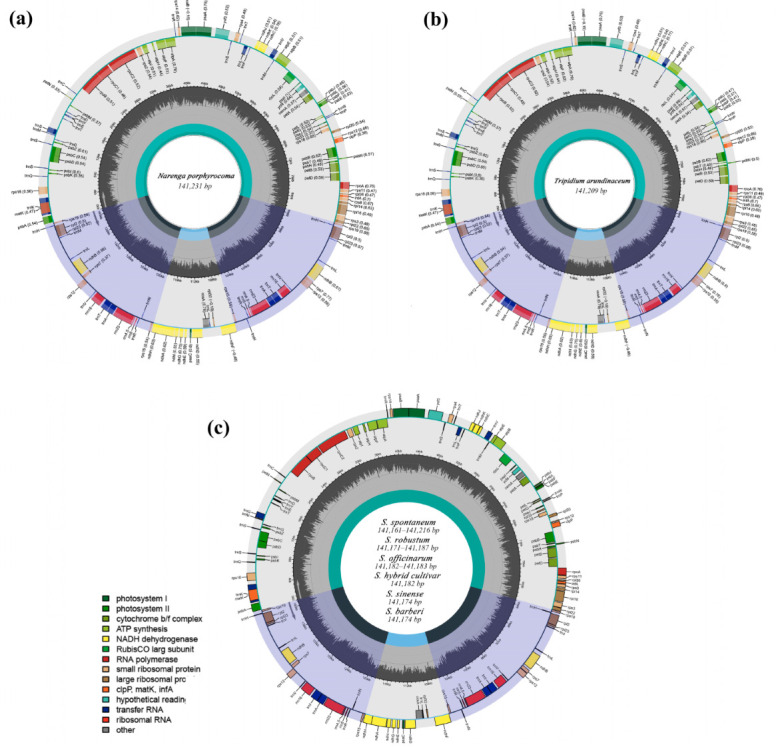
Structure of the assembled and annotated chloroplast genomes. The color of genes indicate their function. The genes presented outside the circle are transcribed counterclockwise, whereas the genes presented inside the circle are transcribed clockwise. The gene content and organization are similar for all *Saccharum* species, and, therefore, one figure (**c**) is drawn as representative of all three species. (**a**) *N. porphyrocoma*; (**b**) *T. arundinaceum*; (**c**) *Saccharum* species.

**Figure 2 ijms-23-07661-f002:**
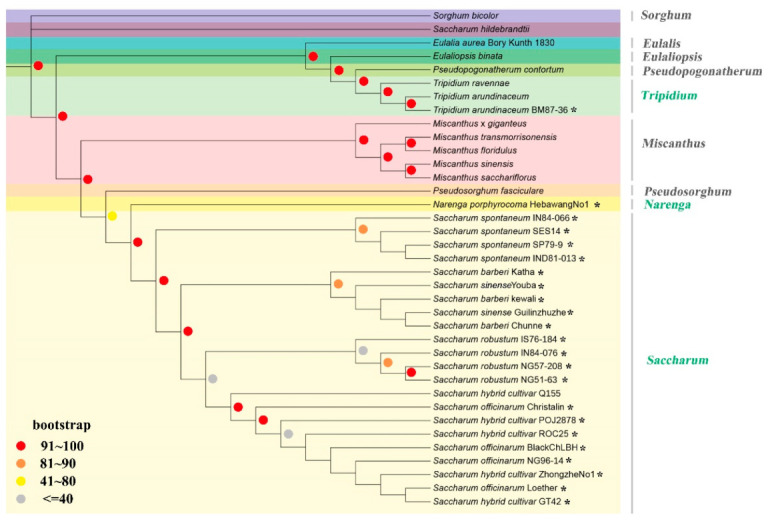
Phylogram based on 74 protein-coding genes shared among Saccharinae species (23 newly assembled in this study, which are marked with ***,** and 14 published accessions of Saccharinae species in NCBI). *Sorghum bicolor* is included as outgroup.

**Figure 3 ijms-23-07661-f003:**
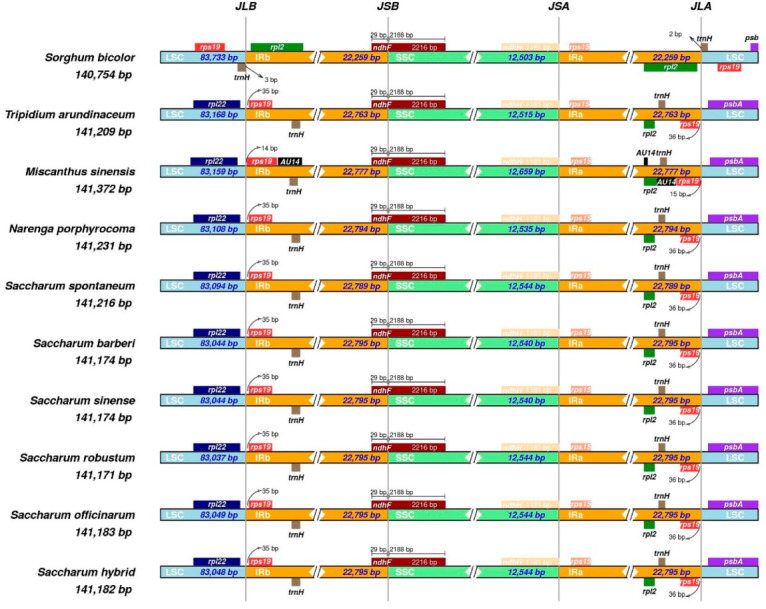
Comparison of border positions of LSC, SSC, and IR regions among nine Saccharinae species and *S. bicolor*. Genes are denoted by boxes, and the gap between the genes and the boundaries are indicated by the number of bases unless the gene coincides with the boundary. Extensions of genes are also indicated above the boxes.

**Figure 4 ijms-23-07661-f004:**
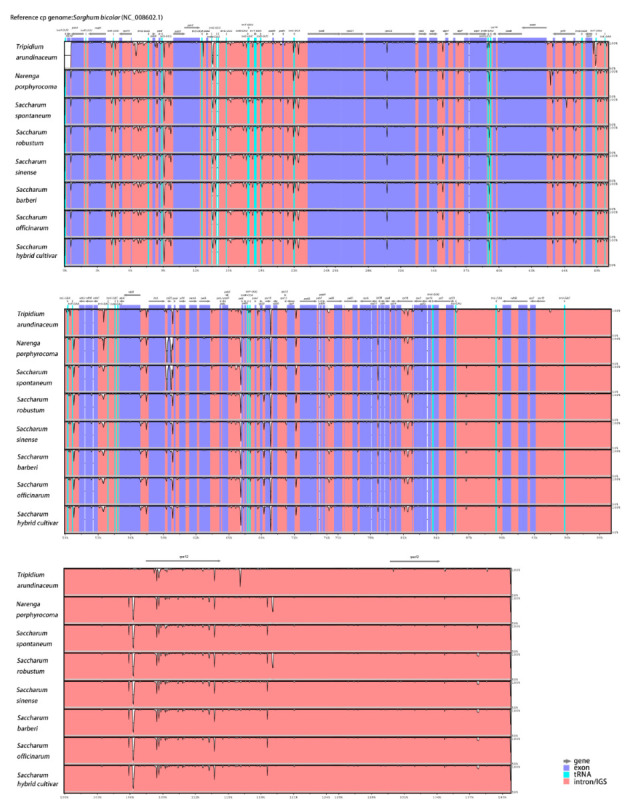
Chloroplast genomes were aligned using the mVISTA program with the annotation of *S. bicolor* as a reference. The X- and Y-scales indicate the coordinates within chloroplast genomes and the percentage of identity (50~100%), respectively. Gray arrows indicate the direction of transcription of each gene. Annotated genes are displayed along the top. The red block is the intron region, the pearlecent Aqua block is the tRNA region, and the purple block is the CDS region.

**Figure 5 ijms-23-07661-f005:**
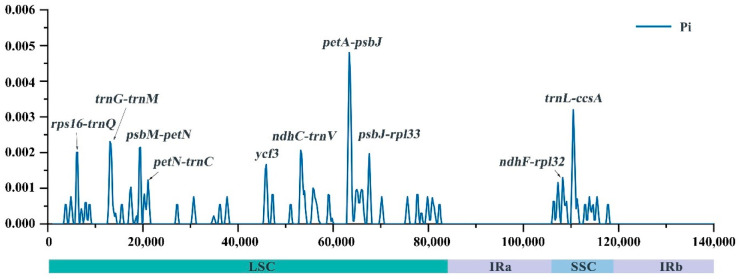
Nucleotide diversity (Pi) based on sliding window analysis in the aligned complete chloroplast genomes of *S. spontaneum*, *S. officinarum*, *S. robustum*, *S. sinense*, *S. barberi*, and *S. hybrids*.

**Figure 6 ijms-23-07661-f006:**
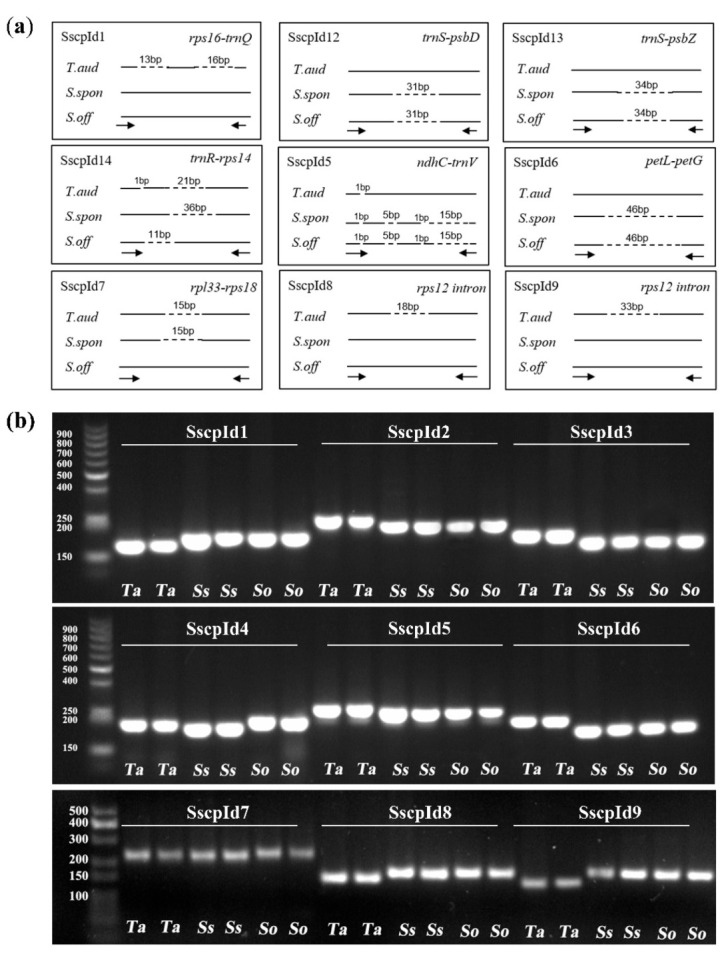
The gel electrophoresis results of the validation of nine molecular markers derived from InDels regions of three chloroplast genomes of *T. arundinaceum*, *S. spontaneum*, and *S. officinarum*. (**a**) The dashed line indicates deleted sequences. Left and right black arrows indicate forward and reverse primers, respectively. (**b**) The gel electrophoresis results from the amplification of InDels markers using designed primers. Abbreviated species names are shown on schematic diagrams, *Ta*, *T. arundinaceum*; *Ss*, *S. spontaneum*; *So*, *S. officinarum*.

**Figure 7 ijms-23-07661-f007:**
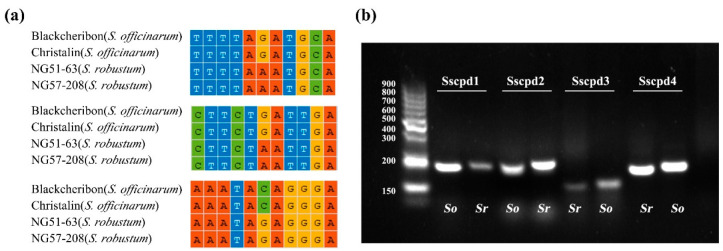
dCAPS makers design based on SNPs between *S. robustrum* and *S. officinarum*. (**a**) The multiple alignments of the chloroplast genomes by MAFFT, (**b**) SNP analysis with dCAPS primers, Sscpd1 and Sscpd2, designed for the SNP site in *trnG-trnM* region, with *TaqI* and *XbaI* restriction sites, respectively. Sscpd3 and Sscpd4 were designed for SNPs in *petA-psbJ* and *rbcL-psaI* intergenic region, with *TaqI* restriction sites. Abbreviated species names are shown on schematic diagrams, *Ss*, *S. spontaneum*, *Sr*, *S. robustum*, *So*, *S. officinarum*.

**Figure 8 ijms-23-07661-f008:**
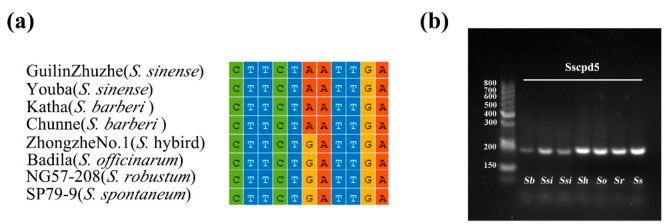
The gel electrophoresis results of the amplification of DNA barcodes using designed primers. (**a**) The multiple alignments of the chloroplast genomes by MAFFT, (**b**) SNP analysis with dCAPS primers, and Sscpd5 designed for the SNP site in rbcL-psaI region with TaqI restriction sites. Abbreviated species names are shown on schematic diagrams, *Sb*, *S. barberi*; *Ssi*, *S. sinense*; *Sh*, *S. hybrid*; *So*, *S. officinarum*; *Sr*, *S. robustum*; *Ss*, *S. spontaneum*.

**Table 1 ijms-23-07661-t001:** The structure and length of the chloroplast genome sequences in *N. porphyrocoma*, *T. arundinaceum*, and *Saccharum* spp. assembled in this study.

Species	Non*-Saccharum*		*S. spontaneum*				*S. robustum*				*S. sinense*	
Accession	BM87-36	HBW1	SES14	SP79-9	IN84-066	IND81-013	NG51-63	NG57-208	IS76-184	IN84-076	Guilinzhuzhe	Youba
Length/bp	141,209	141,231	141,162	141,216	141,191	141,161	141,171	141,171	141,187	141,187	141,174	141,174
LSC/bp	83,167	83,107	83,039	83,093	83,068	83,038	83,036	83,036	83,056	83,056	83,043	83,043
SSC/bp	12,513	12,533	12,542	12,542	12,542	12,542	12,542	12,542	12,538	12,538	12,538	12,538
IR/bp	22,763	22,794	22,789	22,789	22,789	22,789	22,795	22,795	22,795	22,795	22,795	22,795
**Species**	* **S. barberi** *			* **S. officinarum** *				* **S. hybrid** *				
Accession	Katha	Kewali	Chunne	Loether	Christalin	BlackChLBH	NG96-14	POJ2878	ROC25		GT42	ZhongzheNo1
Length/bp	141,174	141,174	141,174	141,183	141,182	141,182	141,183	141,182	141,182		141,182	141,182
LSC/bp	83,043	83,043	83,043	83,048	83,047	83,047	83,048	83,047	83,047		83,047	83,047
SSC/bp	12,538	12,538	12,538	12,542	12,542	12,542	12,542	12,542	12,542		12,542	12,542
IR/bp	22,795	22,795	22,795	22,795	22,795	22,795	22,795	22,795	22,795		22,795	22,795

**Table 2 ijms-23-07661-t002:** Information of nine InDel markers designed for *T. arundinaceum, S. spontaneum*, and *S. officinarum*.

Primer	Forward Reverse	Region	PCR Product Size (bp)		
			T. aru	S. spon	S. off
SscpId1	GGATCGACTCTTTCCCAACA CCCAAAAACGACCAATCTTT	*rps16-trnQ*	130	159	159
SscpId2	TGACAGAGGCAAGAAATAACGA GCCCGCTGAAGAGAAAATC	*trnS-psbD*	235	204	204
SscpId3	AAGACCGGAGCTATCAACCA TGAAAGTGGACTTGCATCTGA	*trnS-psbZ*	157	123	123
SscpId4	CCATTAGACAATGGACGCTTCT TGCTCCATATTCCTTCTTTATGA	*trnR-rps14*	188	172	198
SscpId5	AACCTAATGAAAATCGGATGATTA GCGAATTCCTATTTTGTTTGAA	*ndhC-trnV*	249	228	228
SscpId6	AGGGACTCATGTTCCGTCTG GCATGAAGGGGTTAAATTCC	*petL-petG*	191	145	145
SscpId7	TCGTGTGTTCGATTTTTCCA TTTTGATCCAGAACCAGAAGAA	*rpl33-rps18*	235	235	250
SscpId8	TTTGGCAAAGAAAAATAGAGTACG ATTCCTCAAAACGAGGCTCA	*rps12 intron*	155	173	173
SscpId9	TTTGGCATTAGTTCATGAGGA AAGGAAATGGCTATCAAGAACG	*rps12 intron*	143	176	176

*T. aru*: *T. arundinaceum*; *S. spon*: *S. spontaneum*; *S. off*: *S. officinarum*.

**Table 3 ijms-23-07661-t003:** dCAPS makers design based on SNPs between *S. robustrum* and *S. officinarum*.

Primer	Forward Reverse	Region	Digestion Enzyme	PCR Product Size (bp)/ Target SNP	
				*S. rob*	*S. off*
Sscpd1	CGTACTCCTAATCGAATTTGTATTTT CAGTGGTAAAAGTGTGATTCGTTC	*trnG-trnM*	*Taq*I	226/A	199/G
Sscpd2	CGTACTCCTAATCGAATTTGTATTCTA TGGTAAAAGTGTGATTCGTTCTATT	*trnG-trnM*	*Xba*I	224/A	197/G
Sscpd3	AACAAAAGAATAAATCCAGGGATTCCA CGGGTCCTTACTCCCCTTTA	*rbcL-psaI*	*Taq*I	158/C	174/G
Sscpd4	TCTTCTTTATTCTTCGAATTGATTGC TGCGTATTTGATTCCATTATCGT	*petA-psbJ*	*Taq*I	216/A	201/G

*S. rob*: *S. robustrum*; *S. off*: *S. officinarum*.

**Table 4 ijms-23-07661-t004:** dCAPS maker design based on SNPs between S. sinense/S. barberi and other Saccharum species.

Primer	Forward Reverse	Region	Digestion Enzyme	PCR Product Size (bp)/ Target SNP	
				*S. sin/* *S. bar*	*Other* *S.* spp.
Sscpd5	TCTTCTTTATTCTTCGAATTGATTGC TGCGTATTTGATTCCATTATCGTTCAT	*rbcL-psaI*	*Taq*I	201/A	216/G

*S. sin*: *S. sinense*; *S. bar*: *S. barberi*; *S.* spp.: *Saccharum* spp. (included *Sh*, *S. hybrid*; *So, S. officinarum*; *SR*, *S. robustum*; *Ss, S. spontaneum*.).

## Data Availability

All WGS data used in this work were deposited at National Center for Biotechnology Information (NCBI), BioProject: PRJNA845970; and assembled chloroplast genomes Genbank accession: ON688638-ON688660.

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
