# Peer review of "Comparative Analysis of Chloroplast Genome in Saccharum spp. and Related Members of ‘Saccharum Complex’"

_ijms, 2022, doi:10.3390/ijms23147661_

Round 1
Reviewer 1 Report
Well-written manuscript, methodology consistent with the objectives.
It is an important phylogenomic contribution, but also for the production of variable markers between and within species, which makes the results possible to be used both for evolutionary and commercial studies in this important group of plants.
I did not detect either plagiarism or excessive self-citation.
I think this study is well developed. The methodology for the extraction of DNA, sequencing and phylogenomic analysis is correct. The new markers are important to future studies within the genera. But, as I signed, it is an average interest since no new methodological development exists. On the other hand points to the possibility of interesting population inferences for a crop plant.
Reviewer 2 Report
It's an interesting manuscript and very well written. However, there are a few things that, in my opinion, can and even need to be improved. I present them below:
Abstract
It is too long for over 200 words.
The purpose of the work is missing. It should be corrected according to the journal's guidelines. See intructions for authors. "The abstract should be a single paragraph and should follow the style of structured abstracts, but without headings: 1) Background: Place the question addressed in a broad context and highlight the purpose of the study; 2) Methods: Describe briefly the main methods or treatments applied. Include any relevant preregistration numbers, and species and strains of any animals used. 3) Results: Summarize the article's main findings; and 4) Conclusion: Indicate the main conclusions or interpretations. The abstract should be an objective representation of the article: it must not contain results which are not presented and substantiated in the main text and should not exaggerate the main conclusions. "
There are only results.
Introduction
maybe it is worth adding, how is cp inherited from sugarcane?
Ruler 86
I think it is worth adding that it is the research cited above in the sentence: "The analysis of chloroplasts in Saccharum-related focused because in this manuscript one can think that the purpose of this manuscript is the molecular classification of all Saccharum species and its nearest evolutionary neighbors were elucidated by plastid-based studies ... "
And how to do it - (3) develop efficient markers for parent selection and cross verification in sugarcane breeding program. How can you tell if cp is inherited from the mother? How to verify this cross?
Line 127 (Fig1.c). rather Figure
In section 2.2 Phylogenetic
Line 18. 23 cp genome obtained in this study but in abstract it is 19? then how many were there?
Was the analysis of SNP and InDels carried out on single individuals of each species? Or on a few as sequenced?
line 216 sorghum bicolor as a reference
The nucleotide diversity (Pi) of the chloroplast should also be calculated for individual species because there are several individuals ...
Why some primers are bolded in tables 2 and 3?
Line 320 spp ..
In the M&M section
Again, there is a 16th attempt ... not a 23rd ....
Please specify how many attempts were analyzed in each analysis. because not all clean reads werely aligned
Section 4.6
"To discriminate among the Sacchariane species, specific primers were designed using 489 Primer 3 based on the mutational hotspot regions found in Saccharum and Tripidium arun-490 dinaceum chloroplast genomes." How many species and how many individuals of these species were analyzed. If all, please add that all, if not, which and why were selected.
In conclusions
The authors showed little intraspecies level, maybe it is worth adding the intra and inter variability?
The final sentence is probably over the top: "Molecular markers developed in this study were inexpensive and efficient, and thus were helpful to accelerate the process of marker-assisted breeding in sugarcane for parental selection and cross verification."
How are the markers developed by the authors to accelerate this process and how to verify the cross if only the mother can be identified in this way?
Reviewer 3 Report
The work dealt with the chloroplast genome of the sugarcane and its close relatives. 19 chloroplast genomes had been sequenced and assembled including 3 species of the genera Saccharum (S. robustum, S. sinense, and S. barberi firstly reported in this study), besides Tripidium arundinaceum and Narenga porphyrocoma. The authors presented presented data for the new sequenced chloroplast genome of the species and produced a phylogeny based on their results. The authors also estimated the GC content, and analyzed the structure of the chloroplast genome.
The results are good and well presented in most of the paper. However, I would suggest a revision of the English language. There is also a need to review the abstract, since no methods are described there. It is also important to justify the choice of Sorghum as an outgroup
Some minor changes are suggested in a annotated copy.

Round 2
Reviewer 2 Report
This version of the manuscript is much better. Thanks to the Authors for responding to my comments and making changes.
However, in the Introduction section, I suggest that the Authors in the Introduction section, to balance the high optimism and hopes related to the cp genome analysis, also pay attention to less spectacular applications or examples of works in which this methodology has not been fully proven. The cp analysis of genomes does not always solve the puzzle of complexes of closely related species. It should ring out.
It would be nice to add a reference to such a strong statement - "In addition, the cp genomes have proven to be more informative than cp DNA fragments in revealing the phylogeny of land plants."
In my opinion, it is worth noting that the analysis of cp genomes does not guarantee and does not always allow to unravel the mysteries of plant complexes. I suggest you read the work on the Pinus mugo complex, which would ideally enrich the Introduction or Discussion section - New Insight into Taxonomy of European Mountain Pines, Pinus mugo Complex, Based on Complete Chloroplast Genomes Sequencing (https://doi.org/10.3390/plants10071331).
I think it is worth presenting arguments for and against the given methodology. It is so both more objectively and scientifically. Please consider whether it is worth adding it to the reference to broaden the view on the use of cp genomes in the study of relationships.
The conclusions are more subdued and relevant in this version.
